# Genome-Wide Identification of *NDPK* Family Genes and Expression Analysis under Abiotic Stress in *Brassica napus*

**DOI:** 10.3390/ijms25126795

**Published:** 2024-06-20

**Authors:** Long Wang, Zhi Zhao, Huaxin Li, Damei Pei, Zhen Huang, Hongyan Wang, Lu Xiao

**Affiliations:** 1Academy of Agricultural and Forestry Sciences, Qinghai University, Xining 810016, China; wanglong5898@163.com (L.W.); zhaozhi918@sohu.com (Z.Z.); lhuaxin2000@163.com (H.L.); pdm2022@163.com (D.P.); 2Laboratory for Research and Utilization of Qinghai Tibet Plateau Germplasm Resources, Xining 810016, China; 3Key Laboratory of Spring Rapeseed Genetic Improvement of Qinghai Province, Xining 810016, China; 4Qinghai Spring Rape Engineering Research Center, Xining 810016, China; 5State Key Laboratory of Crop Stress Biology for Arid Areas, College of Agronomy, Northwest A&F University, Xianyang 712100, China; huang_zhen.8@163.com; 6Laboratory of Plant Epigenetics and Evolution, School of Life Science, Liaoning University, Shenyang 110036, China

**Keywords:** *NDPKs*, gene family, evolution, transcriptome, *Brassica napus* L.

## Abstract

The NDPK gene family is an important group of genes in plants, playing a crucial role in regulating energy metabolism, growth, and differentiation, cell signal transduction, and response to abiotic stress. However, our understanding of the NDPK gene family in *Brassica napus* L. remains limited. This paper systematically analyzes the NDPK gene family in *B*. *napus*, particularly focusing on the evolutionary differences within the species. In this study, sixteen, nine, and eight *NDPK* genes were identified in *B*. *napus* and its diploid ancestors, respectively. These genes are not only homologous but also highly similar in their chromosomal locations. Phylogenetic analysis showed that the identified NDPK proteins were divided into four clades, each containing unique motif sequences, with most *NDPKs* experiencing a loss of introns/exons during evolution. Collinearity analysis revealed that the *NDPK* genes underwent whole-genome duplication (WGD) events, resulting in duplicate copies, and most of these duplicate genes were subjected to purifying selection. Cis-acting element analysis identified in the promoters of most *NDPK* genes elements related to a light response, methyl jasmonate response, and abscisic acid response, especially with an increased number of abscisic acid response elements in *B*. *napus*. RNA-Seq results indicated that *NDPK* genes in *B*. *napus* exhibited different expression patterns across various tissues. Further analysis through qRT-PCR revealed that *BnNDPK* genes responded significantly to stress conditions such as salt, drought, and methyl jasmonate. This study enhances our understanding of the NDPK gene family in *B*. *napus*, providing a preliminary theoretical basis for the functional study of *NDPK* genes and offering some references for further revealing the phenomenon of polyploidization in plants.

## 1. Introduction 

Polyploidization is a widespread phenomenon in plant diversification and evolution, and it is a significant mechanism in the formation of angiosperm species [1,2]. During this process, various changes often occur, such as the loss of homologous genes and changes in gene expression patterns [3]. Studies have found that the gene families of species can also undergo a certain degree of contraction and expansion during the process of polyploidization [4]. Transcription factors play a crucial role in plant growth and development [5,6] and in response to biotic and abiotic stress [7]. NDPKs are a relatively conserved class of protein kinases in plants and are ubiquitous in both prokaryotic and eukaryotic organisms. In most eukaryotes, NDPKs exist as hexamers, while in a few prokaryotes, they exist as tetramers. NDPKs have the function of catalyzing the phosphorylation of substrates. They transfer the phosphate group from NTPs to themselves, then transfer the phosphate group to different NDPs, thereby forming new NTPs and maintaining the nucleotide metabolic balance within cells. Additionally, NDPKs can achieve protein conformation changes and activate their enzymatic activity through self-phosphorylation [8]. Research has revealed that NDPK transcription factors play a significant role in regulating plant energy metabolism, growth, and differentiation and cell signal transduction through this phosphorylation action [9,10,11]. Due to their functional importance, they have been extensively studied in various plants, such as *Solanum tuberosum* L. [8], *Arabidopsis thaliana* L. [12], *Nicotiana tabacum* L. [13], and *Oryza sativa* L. [14].

In plants, NDPK kinases were first identified in *Pisum sativum* L., and their presence was subsequently discovered in other plants as well [15]. NDPKs are categorized into four types according to their functions, among which NDPK I, NDPK II, and NDPK III have been more clearly studied. NDPK I is localized in the cytoplasm and nucleus, playing an important role in plant growth and development, hormone response, and abiotic stress response [16]. Studies have found that the NDPK I gene is highly expressed during the germination of ryegrass seeds and fruit development, with the highest expression levels in the apical buds of *N. tabacum*. Under low-temperature stress, the expression of *OsNDPKI* in rice is significantly upregulated, accompanied by an increase in the phosphorylation level of *OsNDPKIV* [17]. NDPK II is mainly located in the chloroplast and acts as a light-regulated protein kinase. Research has shown that under red light induction, the phosphorylation level of *PsNDPK2* in *P*. *sativum* seedlings is upregulated [18]. *A. thaliana AtNDPK2* knockout mutants exhibit a weakened response to red light and difficulty in opening cotyledons, indicating that *AtNDPK2* plays a positive regulatory role in the phytochrome-mediated light signal transduction pathway [19]. Studies also indicate that NDPK II actively participates in the MAPK signaling cascade reaction and auxin regulation pathway [20]. NDPK III is predominantly localized in the chloroplast stroma and the inner membrane of mitochondria, where it plays a crucial role in energy metabolism and the processes of programmed cell death. Studies show that compared to mature tissues, the expression level of *PsNDPK3* in tender tissues is higher at different developmental stages of *P*. *sativum* [21]. Moreover, NDPK III can also participate in the programmed cell death process through interaction with adenylate kinase [22]. Currently, there are relatively fewer research reports on NDPK IV. Given that it contains an endoplasmic reticulum HDEL signal sequence, NDPK IV is speculated to play a role in the endoplasmic reticulum, but its specific mechanism of action remains unclear.

*Brassica napus* L. (*B*. *napus*) is a typical allopolyploid species in the Brassicaceae family and is one of the important oilseed crops. It originated from natural hybridization and polyploidization between *B*. *rapa* (*Brassica rapa* L.) and *B*. *oleracea* (*Brassica oleracea* L.). *B*. *rapa* (genome AA, *n* = 10) and *B*. *oleracea* (genome CC, *n* = 9) provided the necessary genetic material for the formation of *B*. *napus* (genome AACC, *n* = 19). *B*. *rapa* and *B*. *oleracea* are also crucial for studying polyploidization events and the evolution of gene families. With the completion of the genome sequencing of *B*. *napus*, a foundation was laid for its genomic study [23,24,25]. Therefore, this research, utilizing bioinformatics and comparative genomics approaches, identifies the NDPK gene family in *B*. *napus* at the whole-genome level. It also explores its phylogenetic relationships, cis-acting elements, conserved motifs, and collinear relationships and analyzes the expression patterns of *NDPK* genes in different tissues of *B*. *napus* under various stress treatments. Furthermore, it compares the changes in the NDPK gene family during the formation of allopolyploid rapeseed. The findings of this study will contribute to a deeper understanding of the role of *NDPKs* in the growth and development of *B*. *napus* and offer insights into the molecular mechanisms of gene evolution between *B*. *napus* and its diploid ancestors.

## 2. Results 

### 2.1. Identification and Physicochemical Characterization of NDPK Genes

Using the amino acid sequences of NDPKs from *A*. *thaliana*, and employing tools like HMMER and CD-Search along with homological comparisons, we identified sixteen, nine, and eight *NDPK* genes in *B*. *napus* and its diploid ancestors, respectively. Subsequently, we named the 16 genes in *B*. *napus* as *BnANDPK1b* to *BnCNDPK8b* (including A/C subgenomes). The nine genes in *B*. *rapa* were named from *BrNDPK1a* to *BrNDPK9a*, and the eight genes in *B*. *oleracea* were named from *BoNDPK1a* to *BoNDPK8a*. Most of these genes are located in cytoplasm, chloroplasts, and mitochondria, with a few localized to the endoplasmic reticulum (Appendix A). Statistical analysis was conducted on the numbers of amino acids, molecular weights, and isoelectric points of the proteins encoded by these genes. The results showed that the number of amino acids in the NDPK family ranges from 68aa *(BnANDPK3b)* to 438aa *(BnCNDPK5b)*; the molecular weight ranges from 7648.93 Da (*BnANDPK3b*) to 49,400.05 Da (*BnCNDPK5b*); and there is a wide variation in isoelectric points, with *BnCNDPK8b* having an isoelectric point of 10.1 and *BrNDPK6a* having an isoelectric point of 5.76. Most of the proteins (67%) are basic, while a minority (33%) are acidic. Additionally, we found that the majority of NDPK family proteins are unstable.

### 2.2. Chromosomal Localization of the NDPK Gene Family

Using MapGene2chrom, we mapped the NDPK family genes in *B*. *napus* and its diploid ancestors (Figure 1). As shown in the Figure, the *B*. *rapa* genome contains nine *BrNDPK* genes, predominantly distributed on chromosomes Ar08 and Ar09. Similarly, eight *BoNDPK* genes are located in the *B*. *oleracea* genome, with a majority found on chromosomes Co03 and Co09. The *B*. *napus* genome hosts sixteen *BnNDPK* genes, also showing a concentration on chromosomes An08, An09, Cn03, and Co09. Comparing the distribution patterns of the NDPK gene family in the diploid genomes and the *B*. *napus* genome revealed that the distribution patterns of *NDPK* genes on the chromosomes of *B*. *rapa* and *B*. *oleracea* were highly similar to those in the A and C subgenomes of *B*. *napus*. However, compared to the A subgenome, the Cn09 chromosome in the C subgenome might have lost a gene (*BoNDPK6a*) due to incomplete assembly or evolutionary deletion.

### 2.3. Phylogenetic Analysis of the NDPK Gene Family

To reveal the phylogenetic relationships of NDPK proteins in plants, we constructed a phylogenetic tree using the ML method with NDPK protein sequences from nine plant species, including *A. thaliana*, *O. sativa*, *Zea mays* L., *Spinacia oleracea* L., *P. sativum*, *N. tabacum*, *B*. *napus*, *B*. *rapa*, and *B*. *oleracea* as reference sequences (Figure 2). The results indicated that all NDPK proteins were divided into four subfamilies (I~IV), each containing representative AtNDPK proteins. The number of genes in each subfamily varies, with Groups I~III having a similar number of genes and Group IV containing the fewest. Specifically, Group I includes BnANDPK3b, BnANDPK5b, BnANDPK8b, BnCNDPK4b, and BnCNDPK7b; Group II comprises BnANDPK4b, BnANDPK7b, and BnCNDPK3b; Group III consists of BnANDPK1b, BnANDPK2b, BnANDPK9b, BnCNDPK1b, BnCNDPK2b, and BnCNDPK8b; and Group IV contains BnANDPK6b and BnCNDPK5b. In this phylogenetic tree, the monocot plants, *O. sativa* and *Z. mays*, cluster together. Additionally, it was found that NDPK proteins from diploid ancestors clustered together with their homologous NDPK proteins in *B*. *napus*, showing a good evolutionary relationship.

### 2.4. Motif Composition, Conserved Domains, and Gene Structure Analysis of the NDPK Gene Family

To further reveal the evolutionary relationships of *NDPK* genes in *B*. *napus* and its diploid ancestors, we analyzed their conserved motifs (Figure 3a,d). The results showed that the same evolutionary branch exhibited a similar composition of conserved motifs, yet there were certain differences in the numbers and types of conserved motifs between different branches, indicating functional diversification among different family genes. Among these motifs, Motif 1, Motif 2, Motif 3, and Motif 8 are present in most NDPK proteins, suggesting these motifs are more conserved and might be more significant in their evolutionary process. Moreover, Motif 4 and Motif 6 are unique to the NDPK III subfamily, Motif 7 and Motif 9 to the NDPK II subfamily, and Motif 10 exclusively to the NDPK I subfamily, further indicating that these differential motif compositions may serve as distinctive characteristics of each branch and provide clues for their functional diversification. The analysis of conserved domains showed that *BrNDPKs*, *BoNDPKs,* and *BnNDPKs* all contain the hallmark domain of the NDPK gene family (Figure 3b).

The diversity of gene structures is a primary source of evolution within gene families [26]. To explore the structural diversity of the *NDPK* genes, we analyzed the exon–intron structure of the identified *NDPK* genes. As shown in Figure 3c, *NDPK* genes belonging to the same subfamily exhibit similar structures, yet some display notable differences. Specifically, compared to *BoNDPK2a*, *BnCNDPK2b* lacks two exons and introns each; compared to *BrNDPK9a*, *BnANDPK9b* lacks one exon and intron each; compared to *BoNDPK8a*, *BnCNDPK8b* has an additional exon and intron each; and compared to *BoNDPK1a*, *BnCNDPK1b* lacks three exons and introns each. Interestingly, although *BoNDPK5a* and *BnCNDPK5b* have similar gene structures, the latter incorporates a long intron insertion. Additionally, we found that compared to the NDPK III and NDPK IV subfamilies, the NDPK I and NDPK II subfamilies are more conserved in terms of gene structure. These results suggest that during the evolution of the *B*. *napus* NDPK gene family, most lost some introns and exons, and compared to the A subgenome, the C subgenome might have been more susceptible to changes.

### 2.5. Collinearity Analysis of the NDPK Gene Family

Studies have shown that gene duplication is one of the key factors influencing gene family expansion and has significant implications for the evolution of plant genomes [27]. Therefore, we performed intraspecific and interspecific collinearity analysis of the NDPK gene family in *B*. *napus* and its diploid ancestors (Figure 4). The results revealed 18 pairs of collinear genes within *B*. *napus*, with three and four pairs of collinear genes identified in the diploids, respectively. Their interspecific collinearity relationships are the following: there are 27 pairs of collinear genes between *B*. *rapa* and *B*. *napus*, and 23 pairs between *B*. *oleracea* and *B*. *napus*, with most of these relationships being collinear with chromosomes 9 of the A subgenome and chromosome 3 of the C subgenome in *B*. *napus*. There are 12 pairs of collinear genes between *B*. *rapa* and *B*. *oleracea*. The collinearity relationships among *NDPK* genes across different species are all segmental duplications, suggesting that segmental duplication events may be essential in the evolution of the NDPK gene family.

Previous research has indicated that synonymous mutations are not affected by natural selection, whereas nonsynonymous mutations are influenced by natural selection during evolution [28]. The ratio of nonsynonymous to synonymous substitution rates (Ka/Ks) is commonly used to indicate the selective pressure and evolutionary speed of a pair of duplicated genes. A Ka/Ks ratio greater than 1 suggests positive selection during evolution, less than 1 indicates purifying selection, and equal to 1 denotes neutral selection [28]. In this study, most duplicated gene pairs had a Ka/Ks value less than 1 (Appendix A), with only two pairs of duplicated genes having a Ka/Ks ratio greater than 1 (*BrNDPK3a–BnANDPK3b, BoNDPK8a–BnCNDPK8b*), indicating they were under positive selection. 

Further analysis of these two gene pairs revealed differences in sequence length and important motifs, suggesting differential selective pressures on motif elements during polyploidization. For example, *BrNDPK3a* has a sequence length of 505 bp, while *BnANDPK3b* is 457 bp, with differences in Motif 2 and Motif 8, indicating different levels of selective pressure on their motif elements. Similarly, although *BoNDPK8a* and *BnCNDPK8b* both contain two motifs (Motif 2, Motif 4), *BoNDPK8a* has a sequence length of 1523 bp, whereas *BnCNDPK8b*’s length extends to 12,879 bp due to an inserted long intron.

### 2.6. Analysis of Cis-Acting Elements in NDPK Gene Promoters

Cis-acting elements in gene promoter regions can participate in transcriptional regulation, thereby affecting gene expression levels [29]. We analyzed cis-acting elements in the promoter regions of *NDPK* genes in *B*. *napus*, *B*. *rapa*, and *B*. *oleracea*, finding elements associated with plant growth and development, hormone responses, stress responses, and light responses (Figure 5). Specifically, *B*. *napus* has fewer circadian and O_2_ site elements, which are crucial for circadian rhythm and metabolic regulation, respectively, compared to *B*. *rapa* and *B*. *oleracea*. Regarding hormone responses, we identified nine types, including elements for gibberellin, auxin, methyl jasmonate, salicylic acid, and abscisic acid responses. Notably, salicylic acid elements are reduced, while abscisic acid elements are increased in *B*. *napus*, particularly in the *BnANDPK* and *BnCNDPK2b* genes, indicating their potential roles in stress adaptation. Additionally, *B*. *napus* showed a significant increase in defense-related (TC-rich repeats) and a decrease in anaerobic induction (ARE) elements. Most of the cis-acting elements in *B*. *napus* and its ancestors are predominantly linked to light response, suggesting a key regulatory function in light-mediated processes.

### 2.7. KEGG Enrichment Analysis of BnNDPK Genes

KEGG enrichment analysis was performed on *BnNDPK* genes. The results showed that *BnNDPKs* are primarily enriched in their molecular function, catalytic activity, and nucleoside diphosphate kinase activity (Figure 6). Further investigation into genes on significant pathways revealed that most genes are enriched in these pathways, including all members of the NDPK II subfamily (*BnCNDPK3b*, *BnANDPK4b*, and *BnANDPK7b*), the NDPK III subfamily (*BnANDPK1b*, *BnCNDPK1b*, *BnANDPK2b*, *BnCNDPK2b*, *BnCNDPK8b*, and *BnANDPK9b*), and the NDPK IV subfamily (*BnCNDPK5b* and *BnANDPK6b*). However, only *BnCNDPK4b* and *BnANDPK5b* are included in the NDPK I subfamily. These results suggest that only a few members of this family might be required to function, indicating potential functional redundancy.

### 2.8. Expression Pattern Analysis of BnNDPKs Genes

To understand the potential biological functions of *NDPKs* genes in *B*. *napus*, RNA-Seq data from different tissues and developmental stages were used to analyze the expression patterns of *NDPK* genes (Figure 7). As observed, the expression levels of most *NDPK* genes vary greatly across different tissues. Only a few *NDPK* genes expressed (TPM > 60) across different tissues and organs, including *BnANDPK5b*, *BnANDPK8b*, *BnCNDPK4b*, and *BnCNDPK7b*, especially *BnANDPK5b*, which had the highest expression level in its lower stem epidermis (TPM = 272), indicating these genes play significant roles in these tissues (Figure 7a). As shown in Figure 7b, the expression levels of *NDPKs* also vary at different developmental stages within the same tissue. Specifically, the expression levels of *BnANDPK5b*, *BnANDPK8b*, *BnCNDPK4b*, and *BnCNDPK7b* were significant, with higher gene expression levels in the early stages of seed development (30 DAF, TPM > 100), which then tended to decreased over time. Notably, these four genes, all from the NDPK I subfamily, suggested that the NDPK I subfamily may be an important subfamily of the NDPK family and play a key role in the development of *B*. *napus*. As shown in Figure 7c, during the development of siliques, seven genes exhibited high expression levels, originating from the NDPK I subfamily (*BnANDPK5b*, *BnANDPK8b*, *BnCNDPK4b*, *BnCNDPK7b*) and the NDPK III subfamily (*BnANDPK2b*, *BnCNDPK2b*), all with high expression levels (TPM > 40). However, *BnANDPK5b* and *BnCNDPK4b* showed higher expression levels in the early stages of silique development (TPM > 180), suggesting they may be core genes within the NDPK gene family.

### 2.9. Analysis of BnNDPK Gene Expression Patterns under Salt Stress

To elucidate the biological functions of *BnNDPK* genes, their expression patterns under salt stress were analyzed using qRT-PCR (Figure 8). The results indicate that, compared to the control (0 h), genes in the NDPK I subfamily showed similar expression patterns, with most exhibiting a downward trend. Genes in the NDPK II subfamily initially increased and then decreased, reaching their peak at 6 h post-stress. Except for *BnANDPK1b*, genes in the NDPK III subfamily had higher expression levels, showing an initial increase and then a decrease, particularly for *BnANDPK2b*, *BnANDPK9b*, *BnCNDPK1b*, *BnCNDPK2b*, and *BnCNDPK8b*, which peaked at 12 h post-stress. Genes in the NDPK IV subfamily (*BnANDPK6b*, *BnCNDPK5b*) exhibited similar trends, peaking at 12 h post-stress, suggesting their significant roles during salt stress.

### 2.10. Analysis of BnNDPK Gene Expression Patterns under Drought Stress

To further explore the biological functions of *BnNDPK* genes under abiotic stress, drought stress treatments were conducted. The qRT-PCR results (Figure 9) show that, compared to the control (0 h), most genes in the NDPK I subfamily followed an initially increasing and subsequently decreasing pattern; similarly, genes in the NDPK II subfamily showed this trend, with *BnANDPK4b* and *BnCNDPK3b* peaking at 6 h of stress treatment, whereas *BnANDPK7b* peaked at 12 h. Most genes in the NDPK III subfamily exhibited an increasing then decreasing trend, peaking at 12 h, especially *BnANDPK2b*, *BnANDPK9b*, *BnCNDPK1b*, and *BnCNDPK2b*. *BnANDPK6b* and *BnCNDPK5b* from the NDPK IV subfamily showed a similar expression pattern, reaching their highest expression levels at 12 h of stress treatment, indicating that these genes have a positive response to drought stress.

### 2.11. Analysis of BnNDPK Gene Expression Patterns under Plant Hormone Stress

Given the abundance of plant hormone response elements (MeJA) in the promoters of the NDPK gene family, we analyzed their expression patterns under MeJA stress. The qRT-PCR results (Figure 10) indicate that, compared to the control (0 h), most genes in the NDPK I and II subfamilies showed a downregulated expression trend, peaking at 3 h of stress, whereas *BnANDPK3b* and *BnANDPK7b* exhibited different trends, peaking at 9 h. In the NDPK III subfamily, *BnANDPK1b*, *BnCNDPK1b*, and *BnCNDPK8b* peaked at 12 h, while *BnANDPK2b*, *BnANDPK9b*, and *BnCNDPK2b* peaked at 9 h, with most genes showing an increasing then decreasing expression pattern. *BnANDPK6b* and *BnCNDPK5b* from the NDPK IV subfamily exhibited a similar pattern, peaking at 9 h of stress.

## 3. Discussion

Polyploidization is a prevalent phenomenon in the evolutionary history of angiosperms and serves as a significant driving force in species evolution and formation [30]. Plants exhibit certain advantages in agronomic traits and stress responses after experiencing polyploidization. However, during this process, changes may occur in the plant’s genome size, gene structure, and sequence [3]. Research has found that *NDPK* genes are a relatively conserved class of protein kinases in plants, playing crucial roles in maintaining nucleotide metabolism balance within cells and in regulating plant energy metabolism, growth, and differentiation and cell signal transduction [9,10,11]. Due to their functional diversity, they have been extensively studied in plants such as *S. tuberosum*, *A. thaliana*, *N. tabacum*, and *O. sativa*, but there is scant research on the function of *NDPK* genes in *B*. *napus* and its diploid ancestors. Therefore, our identification and analysis of their NDPK gene family will help in understanding the potential biological functions of *NDPK* genes and their role in the formation of *B*. *napus*.

Sixteen, nine, and eight *NDPK* genes were identified in *B*. *napus*, *B*. *rapa*, and *B*. *oleracea*, respectively, with highly similar distribution patterns. The analysis of the physicochemical properties of NDPK proteins showed differences, suggesting their functions within the plant body might also vary. Subsequently, we constructed a phylogenetic tree using NDPK protein sequences from various plants, where each subfamily contained representative AtNDPK proteins, and NDPK proteins from diploid ancestors clustered together with their homologous NDPK proteins in *B*. *napus*, showing a good evolutionary relationship. This indicates these members are homologous in evolution and likely similar in function. NDPK proteins were divided into four subfamilies, each containing similar conserved motif compositions. However, there were differences in the numbers and types of conserved motifs between different subfamilies, especially for Motif 4, Motif 6, Motif 7, Motif 9, and Motif 10. This further illustrates the functional diversity among them and provides clues for their functional differences. Additionally, the predicted subcellular localization results of different subfamily genes are consistent with previous research [21]. Our analysis of the *NDPK* gene structure revealed that, during evolution, the *B*. *napus* NDPK gene family predominantly lost some introns and exons. Compared to the NDPK III and NDPK IV subfamilies, the NDPK I and NDPK II subfamilies have more conserved gene structures. This conservation likely reflects the functional diversity of *NDPK* genes and the variations that occurred during their evolutionary process, suggesting that the functions of genes in the NDPK I and NDPK II subfamilies may also be more conserved. Additionally, we found that the C subgenome in *B*. *napus* may be more susceptible to changes compared to the A subgenome, which could provide insights for future functional studies of the subgenomes in *B*. *napus*.

The cis-acting elements in gene promoter regions can participate in transcriptional regulation, thereby affecting the expression levels of genes [29]. The results showed that the number of cis-acting elements (359) in *B*. *napus* was comparable to that in its diploid ancestors (367). Further analysis revealed that the cis-acting elements related to plant growth and development (circadian, O_2_ site) are significantly reduced in *B*. *napus*. However, the number of cis-acting elements related to abscisic acid (ABRE) in diploid ancestors (30) was significantly lower than that in *B*. *napus* (48), especially for the NDPK III subfamily’s *BnANDPK9b* and *BnCNDPK2b*, which are relatively abundant. Additionally, the significant increase in defense-related cis-acting elements (TC-rich repeats) in *B*. *napus* suggests their potential biological roles. This increase highlights the enriched diversity in *B*. *napus* resulting from hybridization and polyploidization, which potentially offers advantages in stress resistance and adaptability. The prevalence of polyploids indicates the importance of whole-genome duplication in plant evolution and diversification. In the collinearity analysis, we found that most genes in the NDPK gene family were subjected to purifying selection and underwent duplication events approximately 800,000 to 38 million years ago, indicating that *NDPK* may represent an ancient gene family. The expression patterns of *NDPK* genes suggested that *NDPKs* are mainly expressed in cotyledons and stem epidermis, while their expression is absent in other organs at different growth stages, which might be related to changes in their gene structure and conserved motifs. Interestingly, during the development of seeds and siliques, the expression levels of four genes (*BnANDPK5b*, *BnANDPK8b*, *BnCNDPK4b*, and *BnCNDPK7b*) from the NDPK I subfamily were quite significant, indicating that the NDPK I subfamily may be an important subfamily within the NDPK family and could play a key role in plant growth and development, consistent with previous research [17].

Previous studies mainly focused on the role of *NDPK* genes in plant growth and development, but less so under abiotic stress. Further analysis by qRT-PCR revealed that different subfamilies had different expression patterns under salt, drought, and hormone stresses, which may have been related to their involvement in different biological processes or functions. However, genes belonging to the same subfamily showed similar expression patterns, and most genes in the NDPK III and NDPK IV subfamilies were positively correlated. Particularly, members such as *BnANDPK2b*, *BnANDPK9b*, *BnCNDPK1b*, and *BnCNDPK2b* from the NDPK III subfamily, and *BnANDPK6b* and *BnCNDPK5b* from the NDPK IV subfamily, suggested that these genes were likely major contributors to the hormone and stress responses in the *B*. *napus* NDPK gene family. Moreover, phylogenetic analysis has shown that compared to the NDPK I and NDPK II subfamilies, the NDPK III and NDPK IV subfamilies display a closer evolutionary relationship. Studies have shown that drought and salt stress can accelerate the accumulation of ABA in plants, which in turn induces the expression of related genes, thereby conferring stress resistance to plants. Notably, *BnANDPK2b*, *BnANDPK9b*, and *BnCNDPK2b* all contain abundant ABA response elements (ABRE), suggesting they may play a key role in the abscisic acid signaling transduction in *B*. *napus*. Additionally, we also found that *BnANDPK3b*, *BnANDPK6b*, and *BnCNDPK8b* contain higher levels of methyl jasmonate response elements (CGTCA motif and TGACG motif), and compared to the control, their expression is significantly upregulated, indicating they may play a more meaningful role in *B*. *napus*’s response to MeJA stress.

## 4. Materials and Methods

### 4.1. Genome-Wide Identification and NDPK Gene Family Analysis

To explore the NDPK gene family in *B*. *napus* and its ancestors, we retrieved amino acid sequences of NDPK proteins from the *Arabidopsis* database (TAIR) [31]. Additionally, we obtained genome sequences and gene annotations for *B*. *napus*, *B*. *rapa*, and *B*. *oleracea* from the Brassica database (BRAD) [32]. We started our analysis by downloading the NDPK protein domain model (PF00334) from the Pfam database and used HMMER software v3.4 to identify homologous NDPK proteins [33]. To accurately identify *NDPK* genes, we also performed similarity searches using the amino acid sequences of NDPK proteins reported in Arabidopsis (Evalue < 1 × 10^−5^, similarity > 50%). The sequences obtained from the two identification methods mentioned above were retained, and sequences with low coverage were manually removed. Finally, proteins featuring the target domain were kept for subsequent analysis using InterProScan v.5.32-71.0 software and the CD-Search function in NCBI [34,35].

### 4.2. Physicochemical Properties and Phylogenetic Analysis

The obtained candidate NDPK protein sequences were uploaded to Expasy (https://web.expasy.org/protparam/, accessed on 6 March 2024) and WoLF PSORT (https://wolfpsort.hgc.jp/, accessed on 6 March 2024) for analysis of physicochemical properties and subcellular localization prediction (Appendix A) [36,37]. We examined the evolutionary relationships among *NDPK* genes from several species including *A. thaliana*, *O. sativa*, *Z. mays*, *S. oleracea*, *P. sativum*, *N. tabacum*, *B*. *napus*, *B*. *rapa*, and *B*. *oleracea*. We constructed a phylogenetic tree using the maximum likelihood (ML) method, supported by a bootstrap value of 1000, by downloading NDPK protein sequences from these species [38].

### 4.3. Chromosome Localization and Gene Structure Analysis

The MapGene2chrom software v2.1 was used to depict the chromosomal locations of *NDPK* genes in *B*. *napus* and its diploid ancestors. The conserved domains of NDPK protein sequences were analyzed using the CD-Search tool in NCBI [35]; the online tool MEME Suite 5.5.0 was utilized to predict conserved motifs within the NDPK gene family, by setting the maximum number of identified motifs to 10 [39].

### 4.4. Collinearity Analysis and Cis-Acting Element Prediction of the NDPK Gene Family

We conducted a collinearity analysis of *NDPK* genes in *B*. *napus* and its diploid ancestors using MCscanX software (version 1.1.11) to detect gene duplication events. The results were visualized using Advanced Circos [40]. Further, we calculated the Ka, Ks, and Ka/Ks ratios for duplicated gene pairs using the Simple Ka/Ks Calculator in TBtools software (version 1.120) (Appendix A) [41]. Additionally, we analyzed cis-acting elements in the promoter regions (upstream 2000 bp) of NDPK family members using PlantCARE [42].

### 4.5. KEGG Enrichment Analysis of B. napus NDPK Genes

The eggNOG-MAPPER online tool was used to annotate BnNDPK protein sequences. Subsequently, KEGG enrichment analysis was performed using the R package ClusterProfiler (version 4.2.1), and the Cnetplot package (version 4.2.1) was employed to draw the relationship network diagram [43].

### 4.6. Tissue-Specific Expression Analysis of B. napus NDPK Genes

We studied the potential biological functions of the *B*. *napus* NDPK gene family by analyzing the expression patterns of *BnDNPK* genes across various tissues (root, petal, sepal, pollen, filament, cotyledon, stem peel, and vegetative rosette) and developmental stages of seeds and siliques (14 DAF to 60 DAF). RNA-Seq data were obtained from the BnIR database (http://yanglab.hzau.edu.cn/BnIR, accessed on 4 March 2024 [44]. Gene expression data, measured as TPM, were visualized using the Heatmap function in R language (version 4.2.1).

### 4.7. Different Stress Treatments and qRT-PCR Analysis

ZS11 seeds were cultured in soil:vermiculite (3:1) and then placed in a growth chamber with 12 h light/12 h dark conditions at a temperature of 25 ± 1 °C. At four weeks of age, seedlings were subjected to stress treatments with 200 mmol/L NaCl, 10% PEG-6000, and 200 mmol/L methyl jasmonate solution. Leaf tissues were collected at 0, 3, 6, 9, 12, and 24 h after stress treatment, quickly frozen in liquid nitrogen, and stored at −80 °C. Each treatment had three biological replicates.

Total RNA was extracted from the leaves using a plant total RNA extraction kit (Sangon Biotech, Shanghai, China) and reverse transcribed to cDNA using the PrimeScript^TM^ RT Reagent Kit (TaKaRa, Shiga, Japan). Specific primers were designed using NCBI’s Primer-BLAST tool, with *BnActin7* (GeneBank ID: GBEQ01027912.1) used as the internal reference gene for normalizing gene expression changes (Appendix A). 

Fluorescence signals during qPCR were detected with SYBR Green Master Mix (TaKaRa, Japan) using an ABI-7500 fluorescence quantitative system. We conducted three technical replicates per reaction and calculated relative gene expression levels using the 2^−ΔΔCT^ method. The raw qRT-PCR data are documented in Appendix A. The mean (±SE) expression values were calculated from three independent biological replicates and three technical replicates (*, *p* < 0.05; **, *p* < 0.01; ***, *p* < 0.001).

## 5. Conclusions

This study employed bioinformatics and comparative genomics to explore the biological functions and evolutionary characteristics of the NDPK gene family, which is essential for the formation of rapeseed. We identified sixteen *NDPK* genes in *B. napus* and nine and eight in its diploid ancestors, all sharing homology and identical chromosomal locations. Our evolutionary analysis highlights the pronounced conservation of these genes across monocots and dicots, alongside a noted loss of introns and exons in the NDPK gene family of *B. napus*. Collinearity analysis indicated that these genes underwent whole-genome duplication events, leading to duplicate copies predominantly undergoing purifying selection. Analysis of cis-acting elements in the gene promoters revealed elements associated with light, methyl jasmonate, and abscisic acid responses, with a notable increase in abscisic acid response elements in *B. napus*. Transcriptome data and qRT-PCR analysis confirmed similar expression patterns within NDPK subfamilies. This study conducted bioinformatics and comparative genomics analyses of the NDPK gene family in *B*. *napus* and its diploid progenitors, providing insights for a future understanding of the potential biological functions of this gene family and the occurrence of polyploidization events in plants. Additionally, with the continuous improvements in genome sequencing quality, research on the NDPK gene family, especially regarding plant growth and development, responses to abiotic stress, and plant hormone signal transduction, is expected to deepen further.

## Figures and Tables

**Figure 1 ijms-25-06795-f001:**
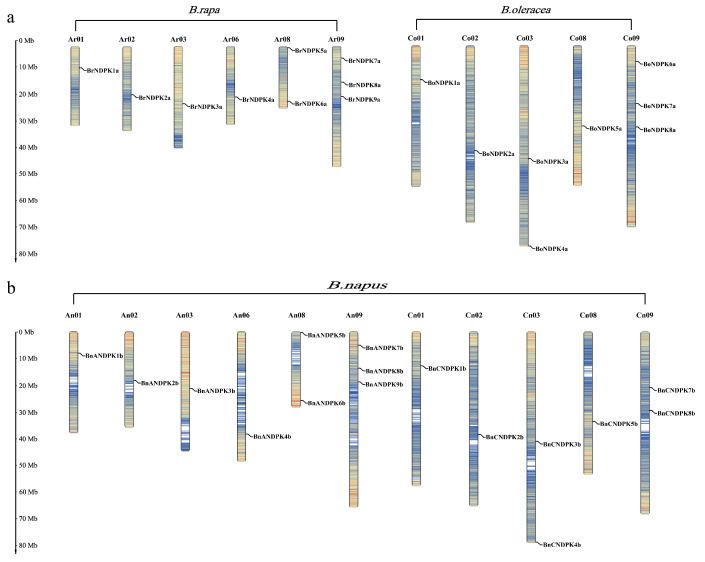
Chromosomal localization analysis of the NDPK gene family on *B. rapa* and *B. oleracea* (**a**) and *B. napus* (**b**).

**Figure 2 ijms-25-06795-f002:**
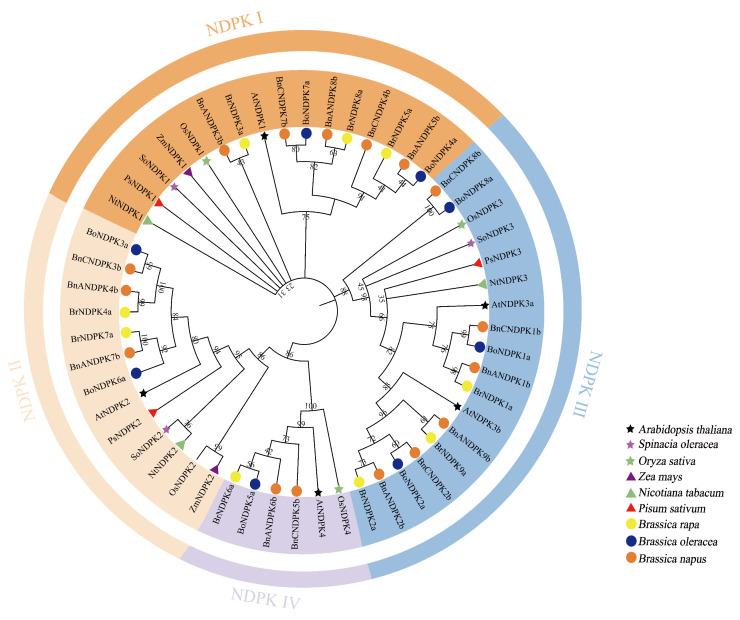
Phylogenetic tree of NDPK proteins from different species. Branches of different colors represent different groups/subfamilies, with orange indicating Group I members, light yellow for Group II, blue for Group III, and purple for Group IV.

**Figure 3 ijms-25-06795-f003:**
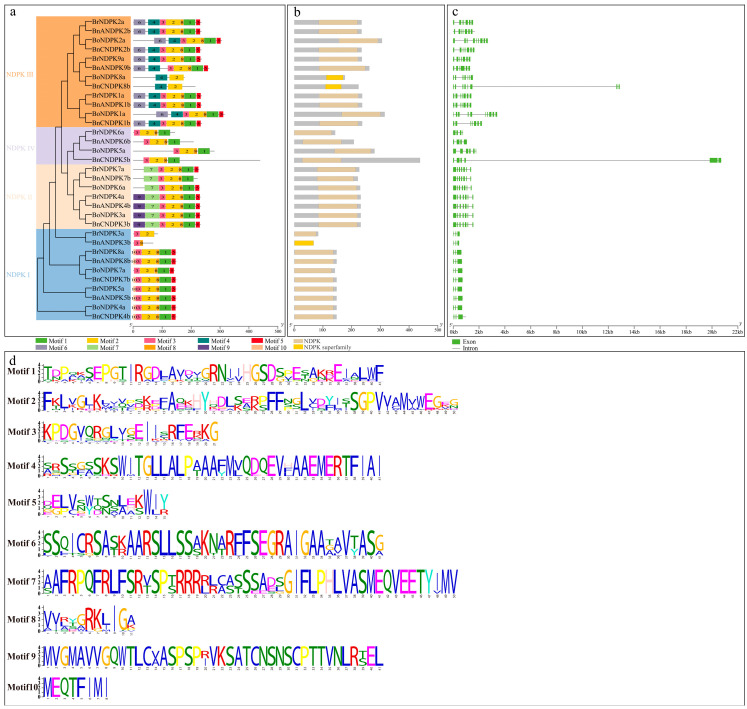
Gene structure and conserved motif analysis of NDPK gene family. (**a**) Conservation motifs; (**b**) conserved domains; (**c**) gene structures; (**d**) logos of each motif of *NDPK* genes in *B*. *napus* and its diploid ancestors.

**Figure 4 ijms-25-06795-f004:**
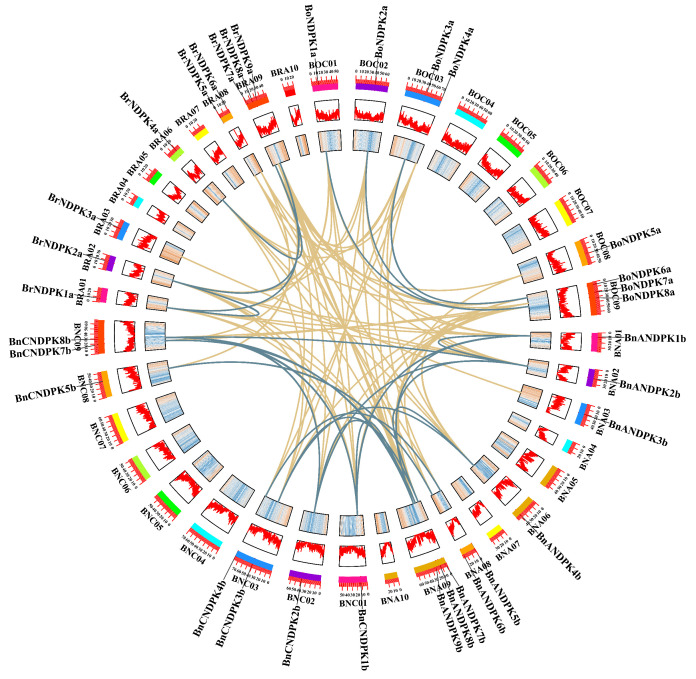
Collinearity analysis of the NDPK gene family in *B*. *napus* and its diploid ancestors, with blue lines indicating intraspecific collinearity relationships and yellow lines indicating interspecific collinearity relationships.

**Figure 5 ijms-25-06795-f005:**
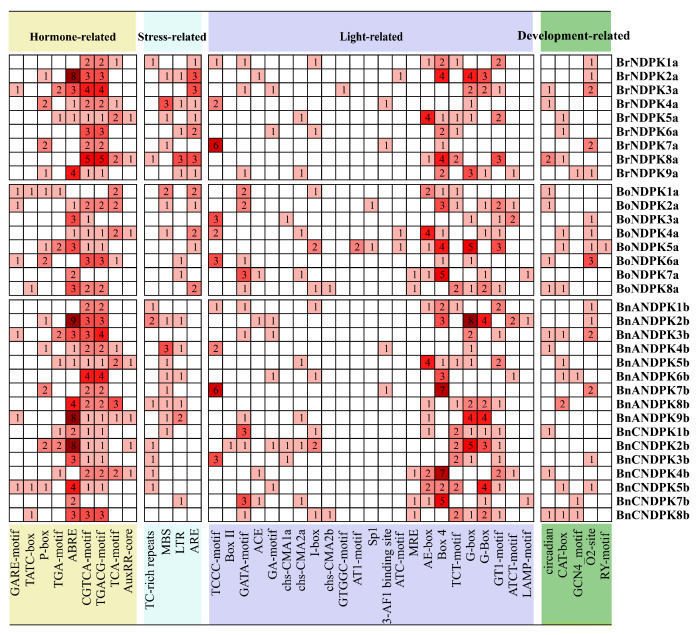
Analysis of cis-acting elements in the promoter regions of *NDPK* genes in *B*. *napus* and its diploid ancestors. Note: Shades of color or size of numbers indicate the number of cis-acting elements.

**Figure 6 ijms-25-06795-f006:**
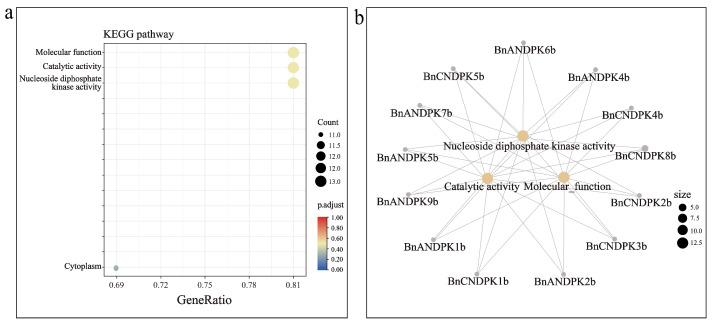
KEGG analysis of the *B*. *napus* NDPK gene family. (**a**) Four pathways enriched in KEGG; (**b**) *NDPK* genes significantly enriched in KEGG. Note: The size of the circle indicates the number of genes enriched into the pathway and the shade of the color indicates the size of the p.adjust.

**Figure 7 ijms-25-06795-f007:**
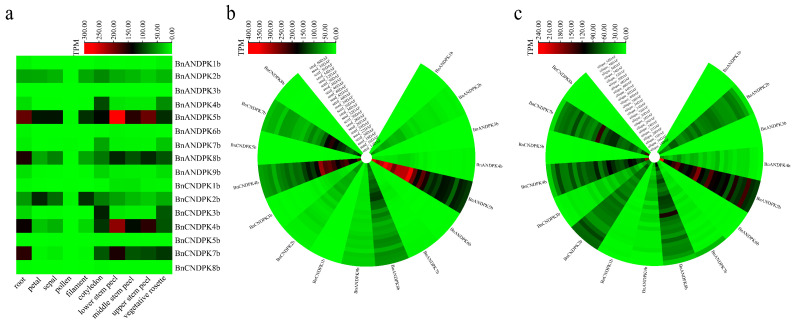
Expression pattern analysis of *BnNDPK* genes in *B*. *napus*. (**a**) Expression patterns of *BnNDPK* genes in different tissues; (**b**) expression patterns of *BnNDPK* genes during seed development from 14 DAF to 60 DAF; (**c**) expression patterns of *BnNDPK* genes during silique development from 14 DAF to 60 DAF.

**Figure 8 ijms-25-06795-f008:**
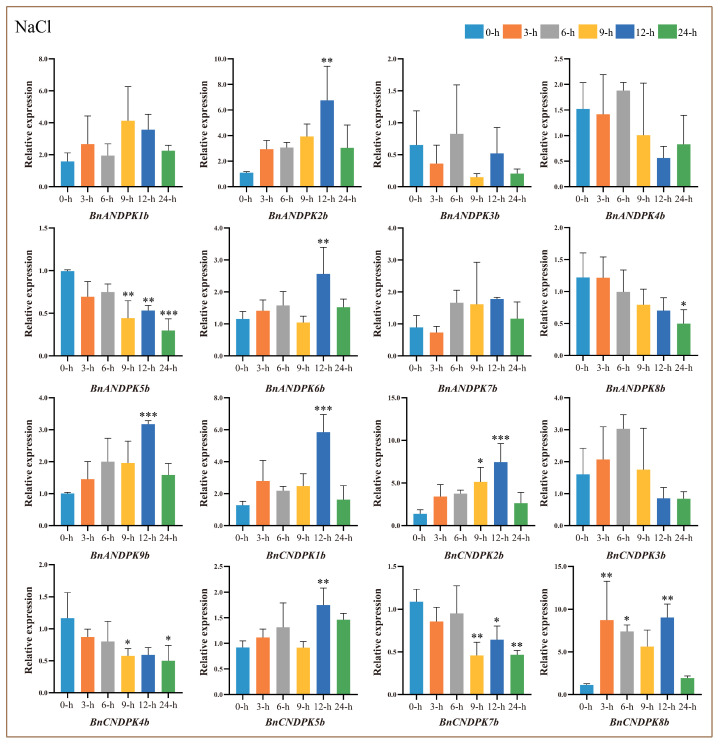
Relative expression levels of *BnNDPK* genes under salt stress at different treatment times (3 h, 6 h, 9 h, 12 h, 24 h) compared to the control (0 h). The mean (±SE) expression values were calculated from three independent biological replicates and three technical replicates (*, *p* < 0.05; **, *p* < 0.01; ***, *p* < 0.001).

**Figure 9 ijms-25-06795-f009:**
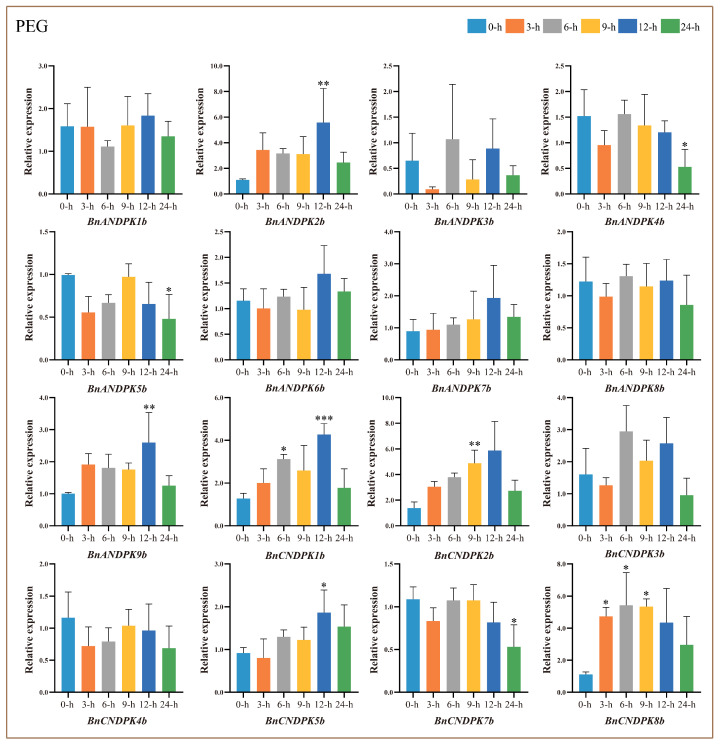
Relative expression levels of *BnNDPK* genes under drought stress at different treatment times (3 h, 6 h, 9 h, 12 h, 24 h) compared to the control (0 h). The mean (±SE) expression values were calculated from three independent biological replicates and three technical replicates (*, *p* < 0.05; **, *p* < 0.01; ***, *p* < 0.001).

**Figure 10 ijms-25-06795-f010:**
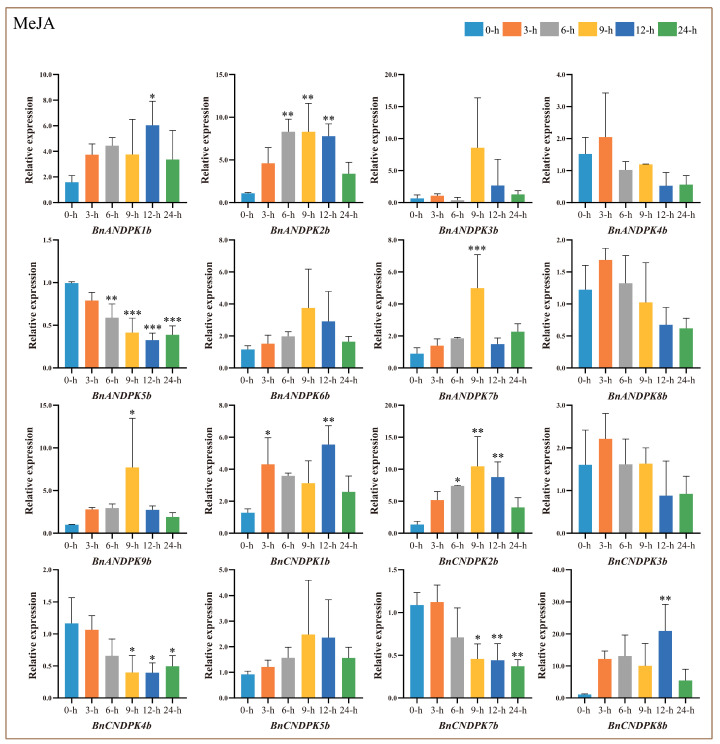
Relative expression levels of *BnNDPK* genes under plant hormone stress at different treatment times (3 h, 6 h, 9 h, 12 h, 24 h) compared to the control (0 h). The mean (±SE) expression values were calculated from three independent biological replicates and three technical replicates (*, *p* < 0.05; **, *p* < 0.01; ***, *p* < 0.001).

## Data Availability

Genomic data were collected from the BRAD database (http://brassicadb.org, accessed on 1 March 2024). Cis-elements were obtained from the PlantCARE database (http://bioinformatics.psb.ugent.be/webtools/plantcare/html/, accessed on 23 March 2024). *Arabidopsis thaliana* sequence information was downloaded from TAIR (https://www.arabidopsis.org/, accessed on 5 March 2024). Transcriptome data were downloaded from (http://yanglab.hzau.edu.cn/BnIR, accessed on 15 March 2024). We used the following online ExPASy website: https://web.expasy.org/protparam/, accessed on 6 March 2024. The Ensembl Plants website used was the following: http://plants.ensembl.org/index.html, accessed on 5 March 2024. All databases in this study are available to the public.

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
