# Peer review of "Genome-Wide Identification of *NDPK* Family Genes and Expression Analysis under Abiotic Stress in *Brassica napus"

_ijms, 2024, doi:10.3390/ijms25126795_

Round 1
Reviewer 1 Report
Comments and Suggestions for Authors
The manuscript is an interesting piece of work on the general picture of Nucleoside diphosphate kinase genes s in Brassica genus.
Some major points needed to be improved to further valorise the manuscript:
- please, check the latin names, some of them are wrong and/or the abbreviation is not used after the first mention
- please check if you are speaking about genes or putative NDPK , sometimes there are confused (such occurring in Table A1)
- very few details are given in Materials and Methods on gene expression (e.g. protocols and reagents)
- no statistical analysis was reported for gene expression analysis
Comments on the Quality of English Languageas reported above
Reviewer 2 Report
Comments and Suggestions for Authors
Comments to authors:
The authors should add graphic abstract.
The authors should do English editing to the manuscript.
Title:
Why you select only Brassica napus and not the other two species, may be you can adjust the title based on that.
Introduction
Please add short note about B. napus, B. rapa, and B. oleracea and their uses and importances.
Material and method
Why you choose 16, 9, and 8 NDPK genes particularly to identify, please explain in more details
Discussion
Please, try to focus on your results analysis and highlight your work importance and its impact.
Conclusion
Future prospective should be added for further development in this area.
Comments on the Quality of English Language
English editing are minor recommended
